# Molecular Mechanisms and Cellular Contribution from Lung Fibrosis to Lung Cancer Development

**DOI:** 10.3390/ijms222212179

**Published:** 2021-11-10

**Authors:** Anna Valeria Samarelli, Valentina Masciale, Beatrice Aramini, Georgina Pamela Coló, Roberto Tonelli, Alessandro Marchioni, Giulia Bruzzi, Filippo Gozzi, Dario Andrisani, Ivana Castaniere, Linda Manicardi, Antonio Moretti, Luca Tabbì, Giorgia Guaitoli, Stefania Cerri, Massimo Dominici, Enrico Clini

**Affiliations:** 1Laboratory of Cell Therapies and Respiratory Medicine, Department of Medical and Surgical Sciences for Children & Adults, University Hospital of Modena and Reggio Emilia, 41100 Modena, Italy; annavaleria.samarelli@unimore.it (A.V.S.); valentina.masciale@unimore.it (V.M.); beatrice.aramini@auslromagna.it (B.A.); roberto.tonelli@me.com (R.T.); marchioni.alessandro@unimore.it (A.M.); giulibru92@gmail.com (G.B.); 72683@studenti.unimore.it (F.G.); darioandrisani@libero.it (D.A.); ivana_castaniere@icloud.com (I.C.); linda.manicardi3@gmail.com (L.M.); antomor93@hotmail.it (A.M.); stefania.cerri@unimore.it (S.C.); massimo.dominici@unimore.it (M.D.); 2Respiratory Diseases Unit, Department of Medical and Surgical Sciences, University Hospital of Modena and Reggio Emilia, University of Modena Reggio Emilia, 41100 Modena, Italy; lucatabbi@gmail.com; 3Oncology Unit, University Hospital of Modena and Reggio Emilia, University of Modena and Reggio Emilia, 41100 Modena, Italy; gioguaitoli@gmail.com; 4Thoracic Surgery Unit, Department of Diagnostic and Specialty Medicine—DIMES of the Alma Mater Studiorum, University of Bologna, G.B. Morgagni—L. Pierantoni Hospital, 34 Carlo Forlanini Street, 47121 Forlì, Italy; 5Laboratorio de Biología del Cáncer INIBIBB-UNS-CONICET-CCT, Bahía Blanca 8000, Argentina; gcolo@inibibb-conicet.gob.ar; 6Clinical and Experimental Medicine PhD Program, University of Modena Reggio Emilia, 41100 Modena, Italy

**Keywords:** idiopathic pulmonary fibrosis, lung cancer, myofibroblast, cancer associated fibroblasts (CAFs), mechanotrasduction

## Abstract

Idiopathic pulmonary fibrosis (IPF) is a chronic, progressive, fibrosing interstitial lung disease (ILD) of unknown aetiology, with a median survival of 2–4 years from the time of diagnosis. Although IPF has unknown aetiology by definition, there have been identified several risks factors increasing the probability of the onset and progression of the disease in IPF patients such as cigarette smoking and environmental risk factors associated with domestic and occupational exposure. Among them, cigarette smoking together with concomitant emphysema might predispose IPF patients to lung cancer (LC), mostly to non-small cell lung cancer (NSCLC), increasing the risk of lung cancer development. To this purpose, IPF and LC share several cellular and molecular processes driving the progression of both pathologies such as fibroblast transition proliferation and activation, endoplasmic reticulum stress, oxidative stress, and many genetic and epigenetic markers that predispose IPF patients to LC development. Nintedanib, a tyrosine–kinase inhibitor, was firstly developed as an anticancer drug and then recognized as an anti-fibrotic agent based on the common target molecular pathway. In this review our aim is to describe the updated studies on common cellular and molecular mechanisms between IPF and lung cancer, knowledge of which might help to find novel therapeutic targets for this disease combination.

## 1. Introduction

Interstitial lung diseases represent a broad spectrum of lung pathologies that affect the lung parenchyma causing diffuse inflammation and fibrosis [1]. Among them, idiopathic pulmonary fibrosis (IPF) is a rare lung disease with an unknown cause and a median survival of 2–4 years from the time of diagnosis. The onset and progression of IPF leads to massive changes in the architecture of lungs and their biomechanical properties that often culminate in respiratory failure due to the impairment of alveolar gas-exchange and the decline of lung functions [2,3]. Although the diagnosis of IPF can be extremely challenging due to the heterogenous nature of this disease [4], it is recognized by clinicopathological criteria, including the radiographic and histological hallmarks pattern of usual interstitial pneumonia (UIP) [5]. Currently, there are two antifibrotic drugs used as a therapeutic strategy for IPF patients, Pirfenidone and Nintedanib, that are able to slow down the respiratory functional decline and improve survival in IPF patients [6,7,8]. Despite this, IPF still has a high mortality rate and the survival times are quite heterogenous [9,10,11]. Although IPF is idiopathic by definition and is not classified as a genetic disease, there are both environmental and genetic risk factors (or “genetic susceptibility”) that can play a fundamental role in the initiation and progression of IPF [12,13]. Among them, the most studied and validated genetic risk factor is represented by the single nucleotide polymorphism in the promoter region of the mucin 5B (MUC5B) responsible for sporadic and familial IPF [12]. Furthermore, among the environmental risks there are exposure to cigarette smoke and inhalation of wood and metal dust, which might severely affect “genetically susceptible” patients [14] with the resulting alteration in the regulation of key genes contributing to the pathogenesis of IPF. Several studies have demonstrated that viral infection, including with the recently prominent SARS-CoV-2, might be responsible for the initiation or exacerbation of pulmonary fibrosis. In particular, it has been postulated that the “cytokine storm” caused by the exaggerated inflammatory response following SARS-CoV-2 infection, as well as micro-thrombotic hypotheses, may predispose patients with COVID-19 pneumonia to aberrant mechanisms of repair and fibrosis [15] culminating in acute lung injury and interstitial lung disease [16]. Although different prospective studies aiming to investigate the long-term pulmonary consequences of COVID-19 are still ongoing, it has been observed in the study of Wu et al. that about 40% of 201 patients with COVID-19 pneumonia developed acute respiratory distress syndrome (ARDS) [17]. Indeed, like SARS-CoV-2, the other two known coronaviruses, both the severe acute respiratory syndrome coronavirus (SARS-CoV; SARS) and Middle East respiratory syndrome coronavirus (MERS-CoV; MERS), caused in some patients interstitial abnormalities and lung functional decline. [18,19].

In the past few years, there has been growing interest in the role of comorbidities in IPF study. Smoking history, elderly age, male sex, and emphysema might represent strong risk factors for developing lung cancer (LC) in IPF patients; thus, besides the concomitant risk factors, IPF itself might be considered a risk factor for lung carcinogenesis [20,21,22]. Lung cancer is the most diffuse cause of cancer death worldwide. It has been estimated that about 85% of LC patients have a diagnosis of non-small cell lung cancer (NSLC). In the majority of patients, the onset of LC is related to tobacco smoking [23] (Figure 1).

Recent studies have focused on the identification of common molecular pathways between IPF and LC for a better therapeutic strategy and optimal management of patients with both diseases. There are many common cellular and molecular mechanisms that might predispose patients to the onset and development of IPF and LC such as the activation and proliferation of both myofibroblasts (IPF) and cancer-associated fibroblasts (CAFs), and the alteration of growth factors expression level [24]. Myofibroblasts represent the cellular key players in IPF since their proliferation and activation under profibrotic stimuli lead to the secretion and deposition of extracellular matrix (ECM) proteins, which are responsible for causing fibrosis [25]. Thus, this process culminates in increased lung structural rigidity that compromises the lung’s biomechanical properties and thickens of the alveolar–capillary barrier with aberrant alveolar gas exchange function. According to recent investigations, the myofibroblasts derive from different lung resident cell populations such as the interstitial lung fibroblasts, the lipofibroblasts [26], lung resident mesenchymal stromal cells (LR-MSCs) [27,28], the perycites [29] and mesothelial cells [30], ruling out the contribution of both epithelial cells in the so-called epithelial mesenchymal transition (EMT) [31] and circulating fibrocytes [32]. In LC, cancer-associated fibroblasts (CAFs) are the cellular key players, like myofibroblasts in IPF pathogenesis [33]. Several studies have been focused on the identification of the origins, biological characteristics, and the role of CAFs as the major component of the stroma during carcinogenesis. Intriguingly, among the components of the tumour microenvironment (TME), CAFs play a fundamental role in drug resistance in NSCLC, protecting tumours from the effects of chemotherapeutic drugs [34,35]. As is the case with IPF, there are several hypotheses still under debate regarding the cellular sources of CAFs in tumours; among them are (1) the resident fibroblasts, which can differentiate to CAFs during the course of tumour progression under the orchestration of specific signalling pathways; (2) cancer associated adipocytes (CAAs) that are present in tumour stroma and might derive from circulating progenitors in the bone marrow [36]; (3) bone-marrow-derived mesenchymal stromal cells (BM-MSCs) and hematopoietic stem cells (HSCs) [37]; (4) epithelial cells through the epithelial mesenchymal transition (EMT) [38]; (5) vascular endothelial cells through the endothelial–mesenchymal transition (EndMT), crucial for tumour angiogenesis [39] (Figure 2) Thus, CAFs exhibit high heterogeneity in terms of origin as well as surface markers and resident organs. During malignancy while the tumour progresses, CAFs contribute to promote tumour growth, metastasis, and drug resistance. Here, tumor cells together with non-malignant stromal cells trigger CAF activation through inflammatory mediators such as transforming growth factor beta (TGF-β), interleukin (IL)-1, and interleukin (IL)-6 that contribute to inflammation and carcinogenesis [40,41]. To this purpose, TGF-β that can be considered as a master molecular regulator of profibrotic signaling, promoting lung cancer progression and triggering mitogenic stimuli to lung cancer cells. Furthermore, among the common signalling pathways characterizing both IPF and LC progression, there is the Wnt/β-catenin pathway that has been involved both in cancer progression and the EMT process through its target genes, cyclin-D1 and matrix metalloproteinase (MMP)-7, contributing to the pathogenesis of IPF [42]. Aberrant activation of phosphoinositide 3-kinase (PI3K)/protein kinase B (AKT) causes cancer invasion and the progression of lung fibrosis [43] with the activation of profibrotic downstream signalling mediators such as TGF-β_1_ and platelet-derived growth factor (PDGF). The sonic hedgehog (Shh) pathway is also activated both in bronchial epithelial cells of honeycomb cysts and in cancer fibroblasts, being responsible for resistance to fibroblast apoptosis, tumour growth, metastasis, and chemotherapy resistance [44]. (Figure 2) Furthermore, cellular senescence is associated with the progression of pulmonary fibrosis through different mechanisms and central players in the lung niche. Among them, Wiley et al. [45] demonstrated that the biologically active profibrotic lipids, namely the leukotrienes (LT), are involved in the senescence-associated secretory phenotype (SASP) which represents one of the mechanisms responsible for the progression of pulmonary fibrosis. They demonstrated that the LT-rich conditioned medium (CM) of senescent lung fibroblasts triggered profibrotic signalling in modified fibroblasts treated with inhibitors of ALOX5, the main enzyme in LT biosynthesis [45]. To this purpose, another recent study from Li et al. [46], stated that blocking the biosynthesis of leukotriene B4 (LTB4) would be an efficient therapeutic strategy in the treatment of both IPF and acute lung injury (ALI). They performed in vivo experiments, where they observed a decreased neutrophilic inflammation in an IPF mouse model at early stage, as well as decreased LPS-induced ALI through LTB4 blocking biosynthesis in vivo. Indeed, several works have been published about the role of LT in lung cancer progression. Among them, Poczobutt et al. [47] showed a selective production of leukotrienes by inflammatory cells of the microenvironment during lung cancer progression through an orthotopic model of lung cancer progression and by liquid chromatograph coupled with tandem mass spectrometry (LC/MS/MS) [47].

Indeed, there are also common genetic mechanisms shared by both diseases based on mutations in surfactant protein genes (SFTPA2-A1) that lead to impaired protein secretion, endoplasmic reticular stress, and apoptosis, culminating in the onset and progression of IPF or adenocarcinoma [24]. (Figure 1) To date, the approved IPF therapies, Pirfenidone and Nintedanib, are also active in LC. In particular, Nintedanib is approved as a treatment in NSCLC, and Pirfenidone has shown anti-neoplastic effects in preclinical studies [24]. Hence, with this review we would like to summarize and clarify the mechanisms implicated in the development of both IPF and LC patients highlighting the similarities in the pathogenesis of both diseases. Hopefully, this would ameliorate the prognosis, opening the possibility of new therapeutic strategies and improving survival among IPF-LC patients.

## 2. Diagnostic Approaches for Idiopathic Pulmonary Fibrosis and Lung Cancer

Idiopathic pulmonary fibrosis (IPF) is a rare pulmonary disease with an incidence ranging from 0.09 to 0.49 in Europe [48]. Among these patients, there is an increased risk of developing lung cancer (LC), with a relative prevalence reported from 2.7% to 48% [49,50]. The correlation between IPF and lung cancer is still being debated [6], although recent studies have demonstrated possible connections [49,50]. In particular, Ozawa et al. [51] showed an increased incidence of lung cancer in a retrospective study of 103 patients with IPF. Similar results have been described by Tomassetti et al. [52], with cancer occurring in 30% of studied patients. One of the most interesting approaches was to analyze the period of onset of lung cancer in IPF-LC patients. In 2021, Alomaish H. et al. [53] reported a LC incidence of 13,5% IPF patients. Because lung cancer has a high incidence in patients with IPF having an important impact on the survival of these patients, the scientific community has focused its attention on identifying predictive factors for lung cancer in IPF patients [54,55,56]. These characteristics are mainly described in older patients with IPF at diagnosis who have a history of smoking and emphysema [51,56,57]. However, the causes inducing lung cancer in this population of patients have not been clarified yet [58,59]. One of the most interesting aspects, which seems important to consider, is a rapid annual decline of forced vital capacity (FVC) as a lung-cancer-predisposing factor [60]. In particular, it seems that tissue damage and abnormal repair is the key to the connections between IPF and LC [59]. Indeed, it has been thought that patients with a very rapid decrease in FVC and consequent IPF progression are more sensitive to lung cancer development; however, further studies will need to be conducted to clarify the common factors between cancer and IPF. A recent study demonstrated a median time from IPF to lung cancer of 38 months [53], although Tomassetti et al. [52] found the time to be around 30 months. Besides the decrease in FVC and carbon monoxide diffusion capacity (DLCO) in patients who developed both IPF and LC, another aspect to investigate are the histopathological findings. Previous studies have shown that squamous cell carcinoma is the most common histologic type encountered in IPF [51,52,53,54,55,56,57,58,59,60,61,62]. This aspect would highlight the importance of considering older patients with a diagnosis of IPF who have a long and heavy history of smoking and whose median FVC and DLCO are slightly lower than normal [63]. In particular, radiological findings are crucial to consider since round or ovoid masses are frequently observed [64,65] in correspondance to the lower lobes [64,66]. The recent guidelines recommend an annual low-dose chest CT scan in high-risk patients [67,68], but there are no recommendations for IPF patients. It is probably important to consider screening patients with IPF, especially in the first two years after diagnosis, with an annual or shorter-term chest CT scan to follow the possible presence and evolution of lung cancer nodules [69]. Since recent studies have described a worse survival rate in patients with IPF and lung cancer [70,71], it would be fundamental to establish a surveillance protocol in order to set a diagnosis and medical treatment for these patients.

The scientific community believes that multidisciplinary approaches are the diagnostic gold standard for patients with moderate IPF and IPF-LC [72,73]. In particular, in patients with IPF of mild to moderate functional impairment (FVC > 50%, DLCO > 35%) and the association of an early-stage lung cancer or metastasis, the common approaches are surgery for the early stages and stereotactic radiotherapy for the advanced stages, together with anti-fibrotic treatments used for more than 50% of patients [74,75,76]. In critical patients with advanced IPF and operable lung cancer but impaired pulmonary function, the recommended treatments are anti-fibrotics, immune checkpoint inhibitors, and targeted therapy [73,77]. Another interesting aspect for diagnostic and therapeutic approaches is the identification of biomarkers, which represents the first step toward future personalized therapies for IPF [73,77]. These approaches may have an important impact in either preventing or monitoring lung cancer in IPF patients. To this purpose, in the last decades, several potential biomarkers have been discovered [78,79].

### 2.1. Diagnostic Biomarkers

The most studied biomarkers able to discriminate IPF from healthy donors are the markers Krebs von den Lungen (KL)-6 [80,81,82,83], the chitinase-like protein (YKL40) [84,85], leucocytes and circulating innate immune cells [78,79,80], and surfactant proteins (SP)-A, -B, -D [86,87,88,89,90,91]. Indeed, the matrix metalloproteinase (MMP7), at higher values, seems to show a higher risk of possible interstitial lung diseases [92]. In particular, MMP7 and MMP1 are important for making a differential diagnosis between IPF and hypersensitivity pneumonitis. The recent PROFILE (Prospective Study of Fibrosis in Lung Endpoints) study that analyzed the molecular profile of more than 100 serum proteins showed a significantly higher level of MMP1, MMP7, and SP-D in IPF patients compared with healthy people. Furthermore, it showed that oncostatin M and cytokeratin 19 fragments (CYFRA-21-1) were markers for IPF patients [54,93,94,95,96,97,98]. These biomarkers may be an important aspect to investigate in the future even for lung cancer. However, the settings of serum biomarkers for clinical perspectives for IPF monitoring and further lung cancer development are still lacking and have not been considered for diagnostic purposes in IPF [99].

### 2.2. Prognostic Biomarkers

Functional decline and respiratory function monitoring are the best approaches at the moment for setting and monitoring IPF progression [96,97], but the possibility has recently arisen of identifying biomarkers with prognostic purposes for IPF. Although the heterogeneity of IPF patients complicates the definition of a univocal approach, a composite scoring system incorporating lung physiology, sex, and age (GAP) is more accurate at predicting mortality [98,99]. These biomarkers seem to improve prognosis for IPF patients [100,101,102,103,104,105,106,107,108,109,110,111,112,113,114]. In particular, a high concentration of MMP7 correlates with IPF severity for the decline of lung function as well as worse survival of IPF [115]. For example, in the PROFILE study, the protein MMP1/8 (CRPM) indicates pulmonary disease progression and very poor overall survival [116]. Another interesting predictor for IPF as well as lung cancer seems to be surfactant protein D and the cancer antigen (CA)19-9 [117].

### 2.3. Radiological Biomarkers

As previously described, several studies defined the high-resolution computed tomography (HRCT) patterns as prospective prognostic biomarkers [118,119]. In particular, a recent study claims that the employment of specific CT-associated tool such as data-driven textural analysis (DTA) [120] associated with a visual and functional changes score, might be useful in predicting IPF progression [120]. However, the definition of a unique method and score set using HRCT characteristics associated with pulmonary function test results is still lacking. Recently, Jacob et al. [121] reported that a computer score set on the quantification of parenchymal patterns including vessel-related structure was able to predict IPF mortality and functional decline, representing a potential non-invasive method for estimating gas-exchange impairment in IPF [121,122].

In summary, keeping in mind the variability and uncertainty of the approaches, the best strategies that need to be considered in asymptomatic middle-aged smokers are: (1) a chest low-dose HRCT or regular CT conducted annually; (2) optimal selection of patients for surgical lung interventions, chemotherapy, and radiotherapy; (3) the role of anti-fibrotics in preventing and treating lung cancer and reducing acute exacerbations of IPF post-operatively; (4) the settings of new molecular biomarkers that may be useful as diagnostic and predictive factors for optimal monitoring of IPF. All these aspects will be better addressed through the setting of a future consensus statement for diagnosis and management of these patients in order to standardize the diagnostic and prognostic approaches as well as also develop more focused medical treatments.

## 3. Common Pathogenic Mechanisms between LC and IPF: Genetic and Epigenetic Alterations in Focus

Several studies going inside the pathogenesis of IPF and LC exhibit great similarity between both diseases concerning the abnormal activation of the signalling pathway, the cellular responses, the activation of lung fibroblasts and their proliferation. To date, there is a growing interest in the epigenetic and genetic abnormalities characterizing IPF and LC that might explain the concomitant manifestation of the two diseases. Although IPF is not considered a genetic disease, genome-wide association analysis (GWA) identified different genetic variants that account for almost 30% of IPF patients that together with the environmental risks play a crucial role in both sporadic and familial IPF [13,123,124]. Among these, the most studied genetic risk factor for both familial and sporadic IPF is the single nucleotide polymorphism rs35705950 in the promoter region of the mucin 5B (MUC5B) gene [12,125]. Normally, in the alveolar epithelium, MUC5B plays a pivotal role together with mucin 5AC (MUC5AC) in the muco-ciliary clearance (MCC), removing inhaled debris and pathogens and contributing to the maintenance of the overall lung homeostasis [126]. To this purpose, it has been shown that the overexpression of full-length murine MUC5B in the AECII of two lines of C57BL/6 mice compromised the muco-ciliary clearance activity, leading to an increase in lung fibrosis of bleomycin-treated mice [1]. Moreover, other genetic variants associated with sporadic and familial IPF are represented by: genes involved in cell–cell adhesion (DSP, DPP9) fundamental in the maintenance of epithelial integrity, genes involved in innate and adaptive immune response (Toll-like receptor signalling, TOLLIP, TLR3), surfactant protein genes (SFTPA2-A1), cytokines and growth factors (IL1RN, IL8, IL4, TGF-β1), genes involved in telomere maintenance (TERT, OBFC1), and cell-cycle regulation genes (KIF15, MAD1L1, CDKN1A, TP53) [127].

Different IPF-related genetic variants have been associated with the risk to develop lung cancer. Among these, there are mutations in surfactant protein genes (SFTPA1, SFTPA2,) that have been studied in lung adenocarcinoma that compromise protein secretion and promote endoplasmic reticular stress and apoptosis [128,129]. In familial IPF, genetic variants of telomerase complex components, such as TERT (telomerase reverse transcriptase) and TERC (telomerase RNA component), lead to shortened telomeres following by genomic instability [130]. To this purpose, several studies suggest a different role of telomerase in IPF and LC since it was found that the expression level of TERT and TERC was significantly lower in the lung tissue of IPF patients compared to NSCLC tissues and controls [131]. Indeed, mutations in the p53 gene that lead to a decrease in the apoptotic process together with mutations in p16, p21, and the Kirsten rat sarcoma virus gene (KRAS) have been found both in IPF and LC [132]. (Figure 1). Furthermore, Maher et al. performed the PROFILE study of a large cohort of IPF patients, four serum biomarkers predictive of disease progression, among which are the cancer-related genes CA-19 and CA-125 [133]. In addition, Allen et al. assessed the genomic profiles of IPF-LC using targeted exome sequencing where they found several somatic mutations, among which were the TP53 and BRAF genes, which were significantly mutated in IPF-LC [134]. Intriguingly, it has been demonstrated that epidermal growth factor receptor (EGFR) and anaplastic lymphoma kinase (ALK) mutations that normally drive the therapeutic decisions in the management of LC patients are either less frequent in IPF-LC adenocarcinoma patients compared with LC adenocarcinoma patients or have not been studied, respectively [135]. The enviromental risk factors, smoking exposure and aging, that might charaterize both IPF and LC patients might induce epigenetic responses such as changes in the methylation patterns that are quite similar between both diseases according to genome-wide methylation analysis [136]. Finally, recent studies show that IPF and LC share the aberrant expression of some microRNA. Among them, miR-21 was both upregulated in patients with IPF, [137] and correlated with poor prognosis in NSCLC patients according to the meta-analysis performed [138]. Finally, miR-29, miR-200, and let-7d were found to be downregulated both in the lung tissue of IPF patients and LC tissues [139,140]. However, it is important to mention that LC is endowed with metastatic potential, disseminating around the body (especially into the brain and bone), unlike IPF, which remains still localized to the lung.

Cigarette smoking together with genetic susceptibility/predisposition and aging can lead to the onset and progression of IPF and LC. Both diseases are characterized by common and similar pathogenic mechanisms. Both in IPF and lung cancer, a massive proliferation of lung resident fibroblasts occurs that contributes to the progression of IPF and lung cancer. In IPF, activated myofibroblasts promote the deposition of ECM leading to an acute exacerbation of the disease, while in lung cancer activated cancer-associated fibroblasts (CAFs) drive the carcinogenesis. Indeed, genetic mutations in surfactant proteins such as SFTPA1 and SFTPA2 have been studied both in familial IPF and lung cancer. Genetic mutations in the p53 gene that lead to a decrease in apoptosis and p16, p21, and KRAS have been found both in IPF and LC.

## 4. The Onset and Progression of IPF: The Leading Role of TGF-β1

According to the most recent studies, IPF is the result of aberrant functional epithelium due to aging and exposure to alveolar injuries together with compromised regenerative capacity of lung tissue that culminate in an imbalance between profibrotic and antifibrotic factors [141,142]. In particular, when injuries occur in normal alveolar lung epithelium, there is a depletion of alveolar epithelial cells 1 (AECI) that are deputed to the alveolar gas exchange functions, being located at the interface with lung vascular endothelium. Here, the depletion of AECI that strongly affects the integrity of lung alveolar epithelium is compensated by the action of alveolar epithelial cells 2 (AECII) that normally secrete the pulmonary surfactant to maintain surface tension, while in this exceptional situation they proliferate and differentiate into AECI. Thus, the AECII are able to ripristinate the integrity of alveolar epithelium during injuries, preserving its functionality [25]. In the lungs of IPF patients the AECII fail to repopulating the alveolar epithelium after their differentiation, compromising both the integrity of alveolar epithelium and the lung’s biomechanical properties. Consequently, this aberrant reparative mechanism of alveolar epithelial cells after injuries marks the onset of IPF and its progression. From here, the AECs enable the disease’s progression through the secretion of profibrotic factors that promote fibroblast migration, proliferation, and differentiation into myofibroblasts. Here, the myofibroblasts cause the distortion of lung architecture with the deposition of exaggerated ECM that increases the stiffening of the lung alveolar epithelium that will translate into impaired elastic properties of lungs and consequent respiratory failure [143].

The profibrotic mediators that contribute to the progression of IPF are represented by transforming growth factor beta-1 (TGF-β1), platelet-derived growth factor (PDGF), tumor necrosis factor (TNF), endothelin-1 (EDN1), connective tissue growth factor (CTGF), osteopontin (OPN), and CXC chemokine ligand 12 (CXCL12). Among them, TGF-β1 represents the master regulator of fibrotic progression since it is responsible for the myofibroblasts’ activity, and the consequent remodeling of ECM [144]. Indeed, the TGF-β that is a chemotactic factor for monocytes and macrophages, and triggers in these cell populations the release of PDGF, IL-1β, basic FGF (bFGF), and TNF-α. To this purpose, an upregulation of the TGF-β expression level has been observed both in AECs and macrophages from the lung tissues of IPF patients [23] as well as in IPF induced in vivo model named the bleomycin mouse model [24]. Indeed, to this purpose the alveolar macrophages play an important role in both IPF and lung cancer progression as highlighted in the work of Jovanovic et al. [145], who found positive membrane PD-L1 expression in alveolar macrophages of IPF lung tissue samples in 9 samples out of 12. They also found elevated concentrations of soluble PD-L1 (sPD-L1) in the serum of IPF patients, significantly higher than a healthy control group. In cancer biology, checkpoint inhibitor agents targeting PD-1/PD-L1 liberate antitumor T cells allowing their activation, proliferation, and killing of tumor cells.

Moreover, the activation of TGF-β receptor complex leads to downstream canonical (SMAD2 and 3) [146] and non-canonical signaling cascades (PI3K, MEK, mTOR, etc.), which modulate the transcription of profibrotic mediators, growth factors, microRNAs, and ECM proteins [147,148]. In particular, activated TGF-β receptors trigger the phosphorylation of SMAD2 and SMAD3, which are translocated into the nucleus for the modulation of transcriptional responses. To this purpose, Zhao et al. demonstrated that knockout mice for SMAD3 reduced bleomycin-induced pulmonary fibrosis in mice [149]. Furthermore, thare are several studies that exhibit cross-talk with TGF-β1 signaling such as the Wnt/β-catenin pathway that might promote epithelial–mesenchymal transition and myofibroblast activation [150]. Recently, it has been demonstrated that the activation of Wnt/β-catenin signaling in the alveolar epithelium of IPF patients might impair lung repair increasing AECII senescence [151]. Indeed, it has been found that Yes-associated protein (YAP) and the transcriptional coactivator with PDZ-binding motif (TAZ) that belong to the Hippo pathway are activated in lung fibrosis. To this purpose, Xu et al. through single-cell RNA sequencing found that the YAP/TAZ, TGF-β, Wnt, and the PI3K axis were activated in the AECs of IPF patients [152,153]. They suggested that mTOR/PI3K/AKT signalling contributed to IPF pathogenesis, both increasing the proliferation of lung epithelial cells and lung fibroblasts. Consequently, TGF-β1 might control the hedgehog pathway (Shh) whose activation in IPF lungs promotes IPF pathogenesis with fibroblast proliferation and aberrant extracellular matrix deposition [36]. Among the other pathways involved, TGF-β1 secretion by the AECs of IPF lungs is due to the activation of αvβ6 integrin that activates the unfolded protein response (UPR), the protective cellular mechanism deputated to the accumulation of aberrant folded proteins [154].

## 5. The Importance of Mechanosignalling in the Progression of Idiopathic Pulmonary Fibrosis

One of the main properties of the lung is represented by the elastic capacity that allows expansion or contraction with inspiratory/expiratory breathing. This particular mechanical capacity is transduced by several mechanosensing proteins to chemical signals into the cells. It is well known that AECII can sense and traduce the breathing mechanism in the alveolar lung epithelium into surfactant secretion. Moreover, stretching might induce the differentiation of AECII to AECI and AECII apoptosis [155,156]. In a fibrotic lung this elastic property is altered due to an increased rigidity in some areas, causing a mechanical disadvantage. We know that the ability of the cell to perceive the extracellular matrix rigidity is due the expression of some proteins called mechanosensors. For example, changes in plasma membrane tension are efficiently detected by mechanosensitive ion-channels [157,158] while mechanical stress between cells is transduced by cadherin-based adhesion complexes [159]. Furthermore, pathological conditions can arise when mechanical parameters in tissues are permanently altered (e.g., tissue stiffening during fibrosis). In addition, the increased ECM stiffness correlates with tumour progression and metastasis [160,161]. Additionally, cell differentiation and migration are induced for substrate stiffness in a phenomenon called durotaxis that is critical for several biological processes like wound healing or cell invasion [162,163]. However, ECM components and rigidity are sensed by cells using a simple receptors and complex transduction machinery. The anchors of this complex machine are the integrins that can connect and transform out-side to inside signalling through the focal adhesion formation and ROCK/myosin activation that can transform the chemical energy (ATP) into mechanical force [164,165]. Integrins connect cells to a wide range of ECM proteins including fibronectin, vitronectin, collagens, and laminins [164]. Mammals express 18 α and 8 β integrin subunits, with which they form 24 distinct α/β heterodimeric integrins that bind specific ECM ligand and signal by assembling hundreds of proteins into a large signalling hub termed adhesome [165,166]. Most cell types express several different integrins, which can synergize to amplify downstream signalling [166,167]. The focal adhesion complex connects integrin signalling to the actin cytoskeleton and gene transcription [168]. This complex transmission system has been found altered in lung fibrosis diseases, while the mechano-interaction between myofibroblast and fibrotic extracellular matrix (stiff matrix) induces a positive feedback loop amplifying lung fibrosis [169]. Even in IPF, fibroblasts display a pathological pattern of several integrins in response to epithelial injury—the expression of α2β1, α3β1, α5β1, α6β4, αvβ5, and αvβ6 are upregulated in epithelial cells [170]. Chen et al. [171] explain that human myofibroblasts expressed high levels of integrin α6 in response to matrix stiffness. Moreover, a stiff matrix induced α6 integrin expression and invasion in lung myofibroblast. The latter was due to RhoA/ROCK and c-Fos/c-Jun transcription complex activation that led to upregulated α6 expression and MMP-2 proteolytical activity associated with an enhanced lung myofibroblast invasion. In addition, mice with conditional α6-gene deletion were shown to be resistant to bleomycin lung fibrosis induction [171,172]. Furthermore, fibroblast cell adhesion to collagen type I is mediated mainly for α2β1 integrin. Xia H. et al. observed that β1 integrin signalling in response to polymerized collagen led to cell proliferation and Akt activation due to low PTEN activity in IPF fibroblasts [173]. Another fibronectin binding integrin, αvβ6, was found elevated in IPF, in response to a profibrotic process that involves TGF-β activation [174,175]. Several studies indicate that inflammation or trauma may increase contractility and induce TGF-β activation, generating an autocrine loop. Whereas TGF-β1 induced αvβ6 integrin via SMAD3, myofibroblast can use αvβ8 integrin to activate TGF-β, generating ECM accumulation and fibrosis [167]. Particularly, knockout mice for αvβ6 and αvβ8 integrins display similar abnormalities to TGF-β1 and TGF-β3 null mice [176]. Furthermore, αvβ6 mice deletion as well as the pharmacological TGF-β inhibition protect from pulmonary oedema formation after bleomycin or endotoxin treatment [177]. Interestingly, it has been recently shown that in a murine bleomycin model, the inhalation of small molecule GSK3008348, an RGD-mimetic, reduced collagen deposition and serum C3M, a marker for disease progression. In particular, it has been shown that GSK3008348 binds to integrin αvβ6, inhibiting the TGF-β downstream profibrotic pathway [178]. As we described before, integrins are involved in the actin cytoskeleton dynamics though GTPases activation. Actin microfilaments exist in a dynamic equilibrium between monomeric G- and polymerized F-actin controlled by the small RhoA-GTPases. Actin cytoskeletal remodelling plays a fundamental role in myocardin-related transcription factor-A (MRTF-A/MLK1/MAL) activation [179]. Hermann et al. demonstrated that fibronectin binding integrins differentially regulate the G- and F-actin ratio though small RhoA activation and induce MRTF-SRF target genes expression [180]. To this purpose, it has been shown that lung fibroblasts in response to matrix stiffening activate nuclear factors that promote myofibroblast differentiation such as F-actin and α-SMA, both over-expressed in a stiff collagen matrix compared with soft and 2 kPa gels. Furthermore, a stiff matrix induces RhoA/ROCK/MLC activation in lung fibroblasts, inducing actin stress fibre formation and lung myofibroblast differentiation [181]. Indeed, ROCK1 and ROCK2 have been the target of studies in pulmonary fibrosis. ROCKs are serine/kinases proteins, and they are the first RhoA mechanical response effectors though myosin light chain (MLC) phosphorylation. The human protein atlas shows elevated ROCK1 protein expression in lung, bronchus, colon, kidney, placenta, spleen, lymph node, tonsil, bone marrow, gallbladder, urinary bladder, and fallopian tube, among others. Instead, ROCK2 is highly expressed in testes and has a medium expression in lungs and other organs [182]. In addition, Knipe and colleagues generated haploinsufficient mice for selective inhibition of ROCK1, ROCK2, or both isoforms. ROCK knockout mice died during embryogenesis, but haploinsufficient mice were viable into adulthood. After bleomycin treatment to induce lung injury, both single- and double-isoform haploinsufficient mice developed a significant reduction in pulmonary fibrosis. Indeed, they observed that ROCK1 significantly protected AEC cells from apoptosis. This study demonstrates that ROCK1 and ROCK2 could protect mice from bleomycin pulmonary fibrosis induction. The specific role of each ROCK isoform involved in lung fibrosis developments is poorly understood. Pharmacological treatment with fasudil in bleomycin-induced pulmonary fibrosis reduced the production of TGF-β1, α-SMA, CTGF, and plasminogen activator inhibitor-1 (PAI-1) [182]. Indeed, a stiff matrix induced the activation of focal adhesion proteins, like FAK, PI3K, RhoA-GTPases, Ras, and YAP, involved in tumour development and metastasis [183,184,185]. Finally, mechanical forces play an important role in fibrosis and cancer progression. Nevertheless, lung cell differentiation during fibrotic tissue stimuli followed by cancer cell transformation and mechanotransduction remains to be elucidated. Further works will need to explain the specific mechanosignalling between fibrotic matrix and lung cell transformation in order to find future therapeutic targets for both diseases.

## 6. The Key Role of Myofibroblasts in Fibrosis-Related Diseases

Myofibroblasts, first identified in granulation tissue during wound healing, are fibroblast-like cells endowed with microfilaments in their cytoplasm similar to that of smooth muscle cells [186]. These cells display contractile properties due to their microfilaments, whose main components are smooth muscle alpha actin (αSMA) and non-muscle myosin type II [187], that make them essential for the regeneration process after wounds [187]. Indeed, myofibroblasts play a pivotal role in the deposition of ECM molecules such as type I collagen, type III collagen, and fibronectin. Myofibroblasts are tightly anchored to the cytoskeleton through integrin-mediated focal adhesions and cadherin-mediated adhesion junctions. Moreover, they express several proteins and collagen cross-linking enzymes such as protein-glutamine gamma-glutamyltransferase 2 (transglutaminase 3), protein-lysine 6-oxidase (LOX), and procollagen-lysine 2-oxoglutarate 5-dioxygenase 2 (PLOD2) [188]. These enzymes reinforce the fibrillar collagen bundles enabling the post-translational modification of collagen molecules during the wound healing. The resulting crosslinks increase the strength of these collagen networks and prevent enzymatic degradation, thereby strengthening the injured tissue [186,187,188,189]. Myofibroblasts can originate in several ways, including from differentiation from fibroblasts, a key process in normal wound healing, orchestrated by growth factors such as TGF-β, WNT, fibronectin cloths, and tissue stiffness [190]. The principal growth factor for myofibroblasts formation is TGF-β, which directly induces extracellular matrix production and αSMA expression. During the fibrotic process, TGF-β modulates several downstream signalling pathways, like SMAD3, PI3K/AKT, and p38 MAPK, promoting the trans-differentiation of myofibroblasts. In particular, the overexpression of SMAD3 enhances both the production of αSMA and the extracellular matrix proteins from fibroblasts [191,192]. Furthermore, the in vivo specific deletion of SMAD3 in mice cardiac fibroblast αSMA production and myofibroblast activation has been demonstrated [193]. Next to these aforementioned stimuli, cellular mechanosensing is another crucial element in the transition of fibroblasts to myofibroblasts. The mechanosensitivity of the cells, described in the previous section, results in activation of various intracellular pathways involving FAK, PI3K/AKT, p38 MAPK, β-catenin, and the activation of transcription activators such as myocardin-like protein 1 (MKL-1), YAP1, and TAZ. Moreover, it has been demonstrated that both MKL-1 and YAP/TAZ directly regulate the myofibroblast phenotype. In particular, the knockdown of MKL-1 lowers αSMA expression in cells grown on a stiff matrix whereas overexpression of a constitutively active form of MKL-1 increases αSMA expression in cells grown on a soft matrix [194,195]. The MLK family, in particular MLK-3, activates type I collagen expression in human embryonic lung fibroblast cell line [10], and interacts with SMAD3 to bind type I collagen and αSMA promoters [196]. Nevertheless, this interplay in the cell line of lung fibroblasts between MKL-1 and SMAD3 can trigger a faster degradation of MKL-1, leading to its suppression [197]. It has been shown that β-catenin serves to counteract this effect of SMAD3 [197], indicating that MKL-1 function depends on the bundling of multiple pathways [197]. Furthermore, the YAP/TAZ pathway can influence matrix stiffness by directly inducing Serpine1 expression. Serpine1 inhibits the activation of plasmin, a protease which degrades extracellular matrix molecules such as fibrin and fibronectin and can activate collagenases, softening the extracellular matrix [195,198]. In turn, YAP/TAZ induces TGF-β and this mechanical activation of TGF-β is enhanced in stiffer matrices [199] (Figure 2). Thus, both YAP/TAZ and TGF-β activity together with matrix stiffness are involved in a sort of feed-forward loop. Finally, another process involved in the transition to myofibroblasts depends on the matrix environment, going beyond the previously described stiffness issue. During tissue repair, fibroblasts that are highly dynamic cells can acquire distinct phenotypes depending on changes in their environment [200] Modulation of the extracellular matrix critically regulates myofibroblast trans-differentiation, the key step in tissue fibrosis. To this purpose, the extradomain-A (ED-A) variant of fibronectin, non-fibrillar collagens, and hyaluronan have been implicated in myofibroblast conversion. In particular, the ED-A fibronectin splice variant is aberrantly expressed during organ fibrosis and it is essential for TGF-β1-mediated myofibroblast conversion [201]. Non-fibrillar collagens (such as collagen VI) have been reported to stimulate myofibroblast trans-differentiation, both in vitro and in vivo [201]. Moreover, the deposition of pericellular hyaluronan may also facilitate and maintain myofibroblast trans-differentiation in response to TGF-β [194], while an independent activation of CD44 signalling by hyaluronan may accentuate the TGF-β/SMAD signalling cascade [24]. Myofibroblasts are also able to modify the biological and mechanical properties of ECM through the secretion of MMPs and tissue inhibitors of metalloproteinases (TIMPs) [202]. Indeed, their contractile capacity results in the structural reorganization of ECM during wound healing and fibrotic conditions [203]. The ECM represents a reservoir of growth factors, creating a highly stimulating environment, which influences cell adhesion, migration, and proliferation [204,205]. Furthermore, the cytoskeleton can regulate trans-differentiation in myofibroblasts and modulate their response to extracellular growth factors, cytokines, and biomechanical stimuli/signals in the ECM [189] Many studies have shown the interplay between myofibroblasts and immune cells, since myofibroblasts can be activated by components of innate and adaptive immunity, and in turn they are capable of modifying immune cells’ behaviour by altering the microenvironment [206,207]. These modifications that cause aberrant ECM production could lead to non-functional fibrotic tissue in multiple organs.

### The Role of CAFs in Tumor Progression

Besides the fibrotic process, the paracrine activity of myofibroblasts, which is carried out with cytokines and growth factors secretion, can affect tumour proliferation and progression, immunosuppression, angiogenesis, and vascular remodelling [208,209]. Cancer associated fibroblasts (CAFs), similarly to myofibroblasts in IPF, are key components of the tumour microenvironment, [210,211], whose recruitment and activation is mainly governed by the cytokines released by cancer cells and infiltrated immune cells. They originate from different cell populations as illustrated in Figure 2 and explained in the Introduction. CAFs which represent the predominant cellular component of the tumour stroma and strongly contribute to the biology of tumour, being essential in the mechanisms of proliferation, invasion, inflammation, angiogenesis, and metastasis that take place in the tumour microenvironment (TME). Furthermore, tumour cells and non-malignant stromal cells secrete different growth factors such as TGF-β1, EGF, vascular endothelial growth factor (VEGF), FGF, TNF-α, IL-1β, and IL-6 [212,213] that promote CAF trans-differentiation and activation, contributing to a pro-inflammatory profile and carcinogenesis. Considering this all, it is possible to note that there is a similarity between fibrosis and LC, and among the action of the different growth factors, TGF-β represents the key driver signalling for the trans-differentiation of both fibroblasts and CAFs, which are essential in tumour progression and resistance to therapies [214,215]. In recent years, many in vitro studies have shown that TGF-β induced EMT into non-tumour epithelial cells though the activation of mTOR and PI3K pathways, leading to the apoptosis resistance of cancer cells, thereby promoting the trans-differentiation of CAFs [216]. Furthermore, CAFs are characterised by high heterogeneity, probably due to the exposure to different tumour secreted factors (TSF), that make them express different molecular markers. Despite their heterogeneity, the molecular mechanisms driving the activation of CAFs and their role in carcinogenesis may be common to various cancers. To this purpose, the main molecular pathways involved besides the TGF-β are the Wnt/β-catenin, the epidermal growth factor receptor, JAK/STAT, and Hippo pathways [217]. Indeed, it has been shown that the different subgroups of CAFs can display opposite biological functions in cancer biology given the identification of some tumour-suppressive CAF populations that are characterized by activated Hedgehog signalling pathways in mouse models of colon, pancreatic, and bladder cancers. [218]. Furthermore, it has been demonstrated the depletion of CAFs in a mouse model of pancreatic ductal adenocarcinoma (PDAC) caused hypoxia, cancer cell proliferation, and cancer progression revealing the tumour-suppressing role of CAFs at this level. [219]. On the other side, Duda et al. [220] demonstrated in preclinical studies of lung cancer that the ablation of CAFs reduced the number of metastases. Indeed, the authors, after analysing human brain metastases from lung, breast, kidney, and endometrium, found high expression of activated CAFs within these metastases. These results demonstrated that the CAFs from the primary tumour might migrate in blood circulation proliferating at the metastatic site. Finally, CAFs contribute to the increased stiffness of a tumour, that leads to the impairment of the blood vessel function, resulting in hypoxia that causes the inefficient uptake of anti-cancer drugs [215,221]. In conclusion, given the pivotal role of myofibroblasts in tumorigenesis, they could represent a promising strategy both to prevent organ fibrosis and to limit tumour progression.

According to the most recent studies, the cell population that gives rise to myofibroblasts in IPF lungs are the interstitial fibroblasts localized close to the vascular endothelium, fundamental for the alveolar gas-exchange function, the lipofibroblasts that are fibroblasts unreached with lipid droplets, the lung resident mesenchymal stromal cells, and the perycites. The pro-fibrotic stimuli lead to an increase in the TGFβ expression level that crosstalks or activates many other downstream pathways (e.g., canonical SMAD3 or non-canonical pathway and growth factors, as listed in the Figure) promoting myofibroblast activation following the aberrant deposition of ECM and progression of IPF. The myofibroblasts’ counterpart in LC, the CAFs, can originate from (1) the resident fibroblasts, (2) cancer associated adipocytes (CAAs) that are present in tumour stroma that might derive from circulating progenitors in the bone marrow [36] (3) bone marrow-derived mesenchymal stromal cells (BM-MSCs) and hematopoietic stem cells [37] (HSCs), (4) epithelial cells through the epithelial mesenchymal transition (EMT) [38], (5) vascular endothelial cells through the endothelial–mesenchymal transition (EndMT). Here, cancer cells which sustain the survival of the tumour secrete different growth factors such as TGF-β1, epidermal growth factor (EGF), vascular endothelial growth factor (VEGF), fibroblast growth factor (FGF), tumour necrosis factor α (TNF-α), interleukin 1β (IL-1β), and interleukin 6 (IL-6) which allow the differentiation of these cell populations into CAFs. Similarly, to IPF, TGF-β, representing the master regulator, activates several downstream signalling pathways. Among them, it induces the epithelial–mesenchymal transition (EMT) into non-tumour epithelial cells though the activation of mTOR and PI3K pathways, leading to the apoptosis resistance of cancer cells, and thereby promoting the trans-differentiation of CAFs. Here, CAFs under the orchestration of specific signalling pathways promote tumour progression.

## 7. Therapeutic Agents for IPF and LC

The many studies and clinical trials related to IPF, investigating the pathogenesis of this disease, have not solved the problem of the survival rate, which is still a dilemma for clinical approaches and therapies [222,223].In this regard, we have summarized in Table 1 the approved therapeutic strategies for both IPF and LC. Concerning IPF, the two antifibrotic drugs Pirfenidone and Nintedanib can both slow down the respiratory functional decline of IPF patients and improve survival in patients according to randomized controlled trials, such as CAPACITY and ASCEND [224]. Nevertheless, IPF still has a high mortality rate. [225,226,227,228]. Pirfenidone has the capacity to inhibit TGF-β, collagen synthesis, and fibroblast proliferation mediating tissue repair [229,230,231,232]. It has been recently shown by Miura et al. [233] that the use of Pirfenidone in IPF patients decreased the incidence of lung cancer compared with IPF patients not treated with the same drug. Furthermore, the scientific community recently demonstrated that Pirfenidone may have an anti-fibrotic role through the Shh pathway [234], which may explain the lower incidence of lung cancer in IPF-treated patients. The introduction of oncological treatments for the control of IPF is not far from becoming reality. In fact, with regard to the connections between lung cancer and IPF, Mediavilla-Varela et al. showed that Pirfenidone synergizes with cisplatin in killing tumor cells and CAFs in a lung cancer preclinical model [235]. Furthermore, inhibitors of vascular receptors, such as VEGF, FGF, and PDGF receptors, namely Nintedanib, have been already approved as specific to IPF [236,237,238]. Moreover, TKIs associated with docetaxel are considered a second-line treatment in NSCLC patients with advanced disease when the patient has been previously treated with standard chemotherapy [239]. In particular, with regard to the common tyrosine-kinase inhibitors (TKIs), such as gefitinib, erlotinib, or afatinib, their role in cancer has been already shown, especially in NSCLC patients, which show mutations activating the epidermal growth factor receptor (EGFR) [138,240,241]. To this purpose, there are ongoing preliminary studies that have identified gefitinib, erlotinib, and afatinib as potential target for IPF treatment. [242]. Indeed, although Imatinib is currently in use for the treatment of NSCLC and was studied as a potential therapy for IPF, there is a strong recommendation against its use in the 2015 IPF Guidelines from ATS/ERS [236]. Indeed, nivolumab, used for lung cancer treatments, which acts on the programmed death ligand 1 (PD-1), has been used in IPF-LC patients. In this regards, Duchemann et al. reported three cases of patients with mild to moderate IPF and lung cancer treated by nivolumab. None of them developed lung toxicity or worsening of IPF on CT during follow-up, and the death was always related to lung cancer progression. They stated that for these three patients with IPF, nivolumab was well tolerated related to their pulmonary condition [237].

A recent class of antifibrotic agents that do not seem to be effective in IPF is the mTOR kinase inhibitors, including everolimus, although this immunosuppressor seems to have some positive effect in advanced lung cancer patients [238,239]. Recently, a new mTOR kinase inhibitor, GSK-2126458, is in a phase I clinical trial for IPF treatment, while it has been studied as a potential therapy forrefractory solid tumours or lymphoma. [243]. A recently approved treatment is rovalpituzumab, currently used in small cell lung cancer (SCLC), although a possible role even in IPF and lung cancer-IPF patients has been noted due to its capacity to interfere and inhibit, in a rat bleomycin model, with the Notch signaling pathway, and hence the improvement of lung fibrosis [244,245,246,247,248,249,250]. Several ongoing clinical trials and preclinical studies (257–29) (Table 2) are currently investigating the role of different molecules in IPF pathogenesis to define their anti-proliferative and anti-fibrotic roles (Table 1) which might have good prospects even against lung cancer; these are the anti-IL-13 antibodies (QAX576 and Lebrikizumab), the anti-CCL2 antibodies (Carlumab and CNTO-888), the anti-TGF-β1 antibodies (Fresolimumab and GC1008), the anti-integrin αvβ6 antibodies (BG0011 and STX-100),and the integrin αvβ6 antagonist drugs (GSK3008348), etc. [251,252,253]. Another drug that is actually used and considered feasible in IPF and lung cancer-IPF patients is vantictumab, which influences Wnt signaling and is in Phase I clinical trials for the role of inhibitor of lung fibrosis [254,255] Indeed, a preclinical study demonstrated the anti-fibrotic effects of CG-745, a histone deacetylase (HDAC) inhibitor, in bleomycin mouse models. They demonstrated that CG-745 prevented collagen synthesis, inflammatory cell accumulation, and cytokine release. The anti-fibrotic effects of CG-745 may suggest a potential therapeutic effect of CG-745 on lung fibrosis [256]. Pre-clinical studies demonstrated that increased expression of HDAC-1 was associated with lung cancer progression. Consequently, treatment with HDAC inhibitors (HDACi) has shown anti-proliferative activity in non-small cell lung cancer (NSCLC) cell lines. Although these are promising results in pre-clinical studies, HDACi has shown only modest efficacy in lung cancer clinical trials [257].

Finally, through a computational analysis, saracatinib has emerged as a promising drug, acting on Src kinases. The clinical trial related to this study (STOP-IPF-Phases 1b/2a) [258] started in October 2020 involving 100 subjects, to explore the application of saracatinib on IPF. This molecular and omics approach allowed the identification of the basaloid cells, a novel and rare cell population detected in the lungs of IPF patients. These cells express some of the common airway basal cell markers (TP63, KRT17, and LAMB3 LAMC2) and SOX9 transcription factor, crucial for airway development and repair, but lack some lung keratins, such as KRT5 and KRT15 [259]. Thus, basaloid cells could represent a novel source of cells to direct the efforts for future target therapies. However, the new medical frontier in the treatment of IPF and lung cancer relies on the use of genomics, which through a disease-specific gene signature allows scientists to identify possible candidates for drug targeting. To date, most approaches to precision medicine have focused on the search for distinct genetic and/or molecular disease subgroups that might be crucial for the management of both familial and sporadic IPF. To this purpose, the analysis of genetic variants performed in the last two decades, together with a recent high-resolution atlas of the resident lung populations, allowed the identification of different genetic mutations that can specifically (e.g., MUC5B, SP-A, SP-d) identify different subsets of IPF patients through a genetic signature/profile [260,261].

## 8. Conclusions

With this review we have shown how IPF might represent a risk factor for developing lung cancer. Both diseases display great similarity concerning the cellular processes and molecular pathways involved in the onset and progression. To date, the therapeutic treament and management of patients for both diseases remains challenging. Although the two approved anti-fibrotic drugs Pirfenidone and Nintedanib increase the survival time of IPF patients and lower the incidence of LC, there is no effective therapeutic treatment preventing the progression of IPF and its related risk of the onset of lung cancer. Nevertheless, the growing interest in investigating the similarity between lung cancer biology and IPF pathogenesis is prompting scientists to test the role of specific target drugs to prevent LC in IPF patients.

## Figures and Tables

**Figure 1 ijms-22-12179-f001:**
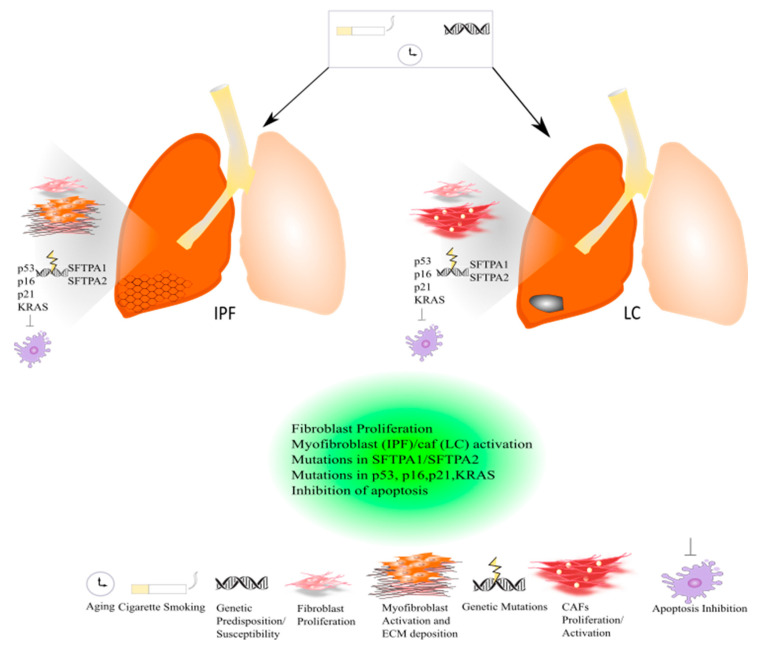
Common pathogenic mechanisms between lung cancer and IPF.

**Figure 2 ijms-22-12179-f002:**
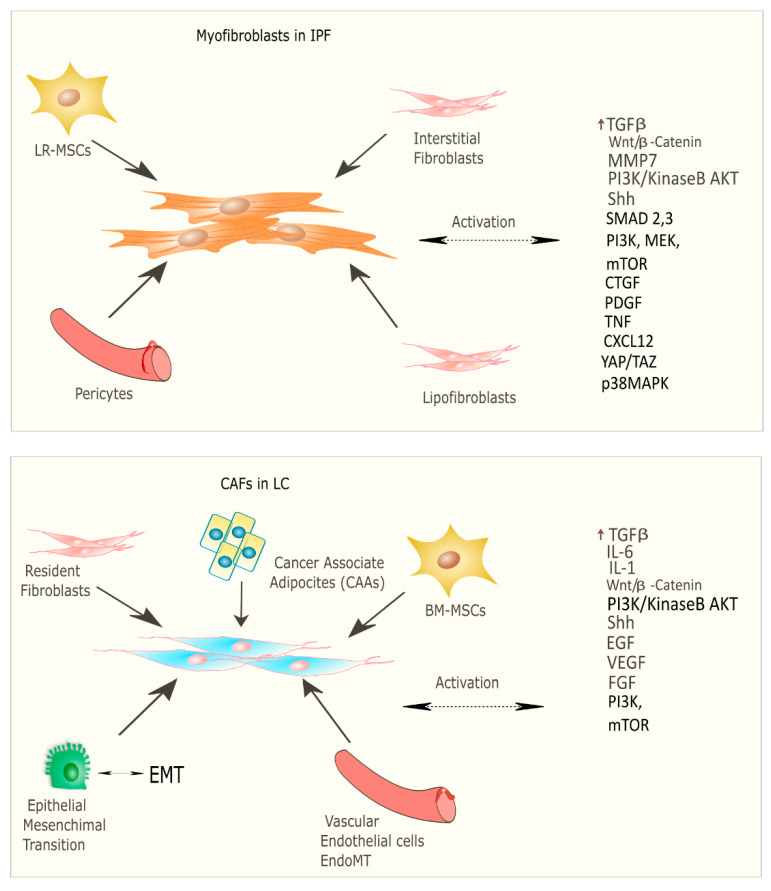
Origin and signalling pathway leading to the activation of myofibroblasts in IPF and CAFs in LC.

**Table 1 ijms-22-12179-t001:** Current approved therapeutic strategies for IPF and LC treatment.

Therapy	Approved for IPF	Approved for LC
Nintedanib	Yes	Yes—In combination with docetaxel (second-line treatment) for ADC-NSCLC
Pirfenidone	Yes	No—Preclinical studies ongoing
Gefitinib	No—Early phase drug discovery study	Yes
Erlotinib	No—Early phase drug discovery study	Yes
Afatinib	No—Early phase drug discovery study	Yes
Imatinib	No—Strong recommendation against its use for IPF patients	Yes
Rovalpituzumab	No—Artesunate (pre-clinical studies)	Yes
Everolimus	No—Not effective in IPF	Yes
Nivolumab	Yes—Used for IPF-LC patients	Yes

**Table 2 ijms-22-12179-t002:** Current phase II–III trials in idiopathic pulmonary fibrosis (IPF).

	Mechanism of Action	Clinical Trial Identifier	Phase of Development	Treatment Duration
PRM-151	Recombinant form of human SAP	NCT02550873	II	28 weeks
Simtuzumab	Anti-LOX antibody	NCT01769196	II	148 weeks
BG00011	Anti-integrin antibody	NCT03573505	II	52 weeks
Tralokinumab	Anti IL-13	NCT01629667	II	52 weeks
Pamrevlumab (FG-3019)	Anti-CTGF antibody	NCT01890265	II	48 weeks
Tipelukast	Leukotriene antagonists	NCT02503657	II	26 weeks
PBI-4050	GPR84 antagonist/GPR40 agonist	NCT02538536	II	20 weeks
Lebrikizumab	Anti IL-13 antibody	NCT01872689	II	52 weeks
KD025	Selective inhibitor of ROCK2	NCT02688647	II	24 weeks
GLPG1690	Autotaxin-LPA inhibitor	NCT02738801	II	12 weeks
CC-90001	Kinase inhibitor targeting JNKs	NCT03142191	II	24 weeks
Rituximab	Antibody targeting CD20	NCT01969409	II	36 weeks
Sirolimus	mTOR inhibitor	NCT01462006	NA	22 weeks
Saracatinib	Src family kinase inhibitor	NCT04598919	IB/2	28 weeks

SAP: serum amyloid P; LOX: lysyl oxidase; IL: interleukin; CTGF: connective tissue growth factor; GPR: G protein-coupled receptor; ROCK: ρ-associated coiled-coil containing protein kinase; JNK: Jun N-terminal kinase; mTOR: mammalian target of rapamycin.

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
