# Peer review of "Molecular Mechanisms and Cellular Contribution from Lung Fibrosis to Lung Cancer Development"

_ijms, 2021, doi:10.3390/ijms222212179_

Round 1

Reviewer 1 Report

Samarelli and collogues discussed the correlation between the development of pulmonary fibrosis and lung cancers with a focus of molecular and cellular mechanisms. The review is comprehensive and provides useful information for readers in the field and has potential fit for the publication in the journal. However, some concerns are also raised as below.

Accumulating studies have demonstrated that viral infection including SARS-COV-2 can initiate or exacerbate pulmonary fibrosis, which was not mentioned in this context.

Many studies have shown that leukotrienes are central mediators for pulmonary fibrosis, and how it may also contribute to the correlation of IPF and lung cancer was not discussed in the manuscript.

Lines 333-335: Macrophages play pivotal roles in IPF and lung cancers, however, the authors only wrote two sentences on it.

Line 119: the introduce of honeycomb cysts in IPF lung was abrupt.

Line 43: abbreviations (e.g. CAF) should be spelled out for its first appearance.

The manuscript would benefit from grammar/expression editing from native speakers.

Author Response

On behalf of all the Authors we would like to thank all the Reviewers for their valuable time and useful contribution to our Manuscript “Molecular mechanisms and cellular contribution from lung fibrosis to lung cancer development”. We strongly appreciate the suggestions and inputs they have given that will definitely improve our manuscript updating the molecular mechanisms that characterize both IPF and lung cancer onset and progression. Following their suggestions we have reorganized the manuscript to better and clearly explained concepts such as the role of Cancer Associated Fibroblasts (CAFs) particularly in cancer biology as well as the description of the current used drugs for IPF and LC. Please find below the details.

Reviewer 1:

Samarelli and collogues discussed the correlation between the development of pulmonary fibrosis and lung cancers with a focus of molecular and cellular mechanisms. The review is comprehensive and provides useful information for readers in the field and has potential fit for the publication in the journal. However, some concerns are also raised as below.

  • Accumulating studies have demonstrated that viral infection including SARS-COV-2 can initiate or exacerbate pulmonary fibrosis, which was not mentioned in this context.

1) We agree with the Reviewer and we have included details on pulmonary fibrosis secondary to viral infection such as SARS-COV-2 and others virus belonging to Coronavirus family such as SARS-CoV and MERS-CoV responsible for the SARS and MERS diseases respectively. Details on page 2, line 68:

“Several studies have demonstrated that viral infection including the recent SARS-CoV-2 might be responsible for the initiation or exacerbation of pulmonary fibrosis. In particular, it has been postulated that the “cytokine storm” caused by the exaggerated inflammatory response following the SARS-CoV-2 infection, as well as micro-thrombotic hypotheses may predispose patients with COVID-19 pneumonia to aberrant mechanisms of repair and fibrosis (15), culminating in acute lung injury and interstitial lung disease (16). Although different prospective studies aiming to investigate about the long-term pulmonary consequences of COVID-19 are still ongoing, it has been observed in the study of Wu et al. that about 40% out of 201 patients with COVID-19 pneumonia developed acute respiratory distress syndrome (ARDS) (17). Indeed, likewise SARS-CoV-2, the other two known coronavirus, both the severe acute respiratory syndrome coronavirus (SARS-CoV; SARS) and Middle East respiratory syndrome coronavirus (MERS-CoV; MERS), caused in some patients interstitial abnormalities and lung functional decline (18) (19).

Many studies have shown that leukotrienes are central mediators for pulmonary fibrosis, and how it may also contribute to the correlation of IPF and lung cancer was not discussed in the manuscript.

2)Agreed. We have included in the manuscript the role of leukotrienes as mediators for pulmonary fibrosis progression and their correlation with both IPF and lung cancer development on page 3 line 146 as followed:

“Cellular senescence is associated with the progression of pulmonary fibrosis through different mechanisms and central players in the lung niche. Among them, Wiley et al; (45) demonstrated that the biologically active profibrotic lipids, namely the leukotrienes (LT), are involved in the senescence-associated secretory phenotype (SASP) which represents one of the mechanism responsible for the progression of pulmonary fibrosis. They demonstrated that the LT-rich conditioned medium (CM) of senescent lung fibroblasts triggered profibrotic signaling in modified fibroblasts treated with inhibitors of ALOX5, the main enzyme in LT biosynthesis (45). To this purpose, another recent study from Li et al; (46) stated that blocking the biosynthesis of Leukotriene B4 (LTB4) would be an efficient therapeutic strategy in the treatment of both IPF and Acute Lung Injury (ALI). They performed in vivo experiments, where they observed a decreased neutrophilic inflammation in an IPF mouse model at early stage, as well as decreased LPS-induced ALI through LTB4 blocking biosynthesis in vivo. Indeed, there have been published several works about the role of LT in lung cancer progression. Among them, Poczobutt et al. showed through an orthotopic model of lung cancer progression and by liquid chromatography coupled with tandem mass spectrometry (LC/MS/MS) selective production of leukotrienes by inflammatory cells of the microenvironment during lung cancer progression (47)”

3) Lines 333-335: Macrophages play pivotal roles in IPF and lung cancers, however, the authors only wrote two sentences on it.

3) We thank the Reviewer for the suggestions and we added some details concerning the role of alveolar macrophages in IPF on page 9 Line 385. Line 385:

To this purpose the alveolar macrophages play an important role in both IPF and lung cancer progression as highlighted in the work from Jovanovic et al (156) where they found positive membrane PD-L1 expression in alveolar macrophages of IPF lung tissue samples in 9 samples out of 12. They also found elevated concentrations of soluble PD-L1 (sPD-L1) in the serum of IPF patients significantly higher compared with healthy control group. In cancer biology, Checkpoint inhibitor agents targeting PD-1/PD-L1 liberate antitumor T cells allowing their activation, proliferation and killing tumor cells.

4) Line 119: the introduce of honeycomb cysts in IPF lung was abrupt.

4) Agreed. We have modified the sentence as followed: “The sonic hedgehog (Shh) pathway is also activated both in bronchial epithelial cells of honeycomb cysts and in cancer fibroblasts being responsible for resistance to fibroblast apoptosis, tumor growth, metastasis, and chemotherapy resistance (39)”. (Figure 2)

5) Line 43: abbreviations (e.g. CAF) should be spelled out for its first appearance.

5)We thank the Reviewer for the suggestion. We have correct accordingly.

6) The manuscript would benefit from grammar/expression editing from native speakers.

6)We thank the Reviewer for the suggestion and we revised the English grammar/expression throughout the Manuscript.

Reviewer 2 Report

Dear authors, 

I reviewed your review manuscript with high interest. I feel that this is a topic that it is not widely study and that it of big interest in the clinics. 

I miss in the text (chapter 6: the key role of myofibroblasts in fibrosis-related diseases and cancer) and in figure 2 the explanation of how other cells are also involve on the process, such as interstitial macrophages and other endothelial/epithelial cells that in my point of view also they play an important role.

It would be great to include a table summarizing the therapeutic agents for IPF and LC (chapter 7), this will make the review more attractive and easy to read and condensate all the information.

Author Response

On behalf of all the Authors we would like to thank all the Reviewers for their valuable time and useful contribution to our Manuscript “Molecular mechanisms and cellular contribution from lung fibrosis to lung cancer development”. We strongly appreciate the suggestions and inputs they have given that will definitely improve our manuscript updating the molecular mechanisms that characterize both IPF and lung cancer onset and progression. Following their suggestions we have reorganized the manuscript to better and clearly explained concepts such as the role of Cancer Associated Fibroblasts (CAFs) particularly in cancer biology as well as the description of the current used drugs for IPF and LC. Please find below the details.

I reviewed your review manuscript with high interest. I feel that this is a topic that it is not widely study and that it of big interest in the clinics. 

1)I miss in the text (chapter 6: the key role of myofibroblasts in fibrosis-related diseases and cancer) and in figure 2 the explanation of how other cells are also involved on the process, such as interstitial macrophages and other endothelial/epithelial cells that in my point of view also they play an important role.

1) We thank the Reviewer 2 for pointing out this issue as we put the reference to figure 2 in an incorrect position (page 12 Line 558) and this created misunderstandings in the text. In fact, placing the reference to figure 2 in the correct position within the text (page 3 line 121, 139, pag 11 line 542 ), it explains the cellular origin of both myofibroblasts in idiopathic pulmonary fibrosis and CAFs in cancer. In this regard, the contribution of both epithelial cells and endothelial cells in giving rise to CAFs is illustrated in Figure 2 and cited on page pag 11 line 542. Indeed, alveolar macrophages play a pivotal role in the progression of both IPF and lung cancer as further discussed on page 8 Line 374-381 in response to Reviewer 1.

2)It would be great to include a table summarizing the therapeutic agents for IPF and LC (chapter 7), this will make the review more attractive and easy to read and condensate all the information.

2)Agreed. Thanks to Reviewer’s suggestion we have reorganized this section (pag 15-16) adding a new Table 1 (pag 17-18) where are listed the current approved drugs for IPF and LC, followed by the previous Table 1 that in the new version of the Manuscript become Table 2 that summarizes the current Clinical PhaseII/III trials for IPF and eventually for LC treatment.

Page 17: Table 1. Current approved therapeutic strategies for IPF and LC treatment

Therapy

Approved for IPF

Approved for LC

Nintedanib

Yes

Yes- in combination with docetaxel (second-line treatment) for ADC-NSCLC

Pirfenidone

Yes

No-

Preclinical studies ongoing

Gefitinib

No-

early phase drug discovery study

Yes

Erlotinib

No-

early phase drug discovery study

Yes

Afatinib

No-

early phase drug discovery study

Yes

Imatinib

No-

Strong reccomandation against its use for IPF patients

Yes

Rovalpituzumab

No-

Artesunate (pre-clinical studies)

Yes

Everolimus

No-

Not effective in IPF

Yes

Nivolumab

Yes-

used for IPF-LC patients

Yes

Reviewer 3 Report

This review paper discusses the molecular mechanisms and cellular contribution from lung fibrosis to lung cancer development. Altohugh the Authors postulated that the correlation between IPF and lung cancer is still being debated, in this review presented a lot of news and reports indicating the association of IPF with lung cancer development. This review give a summary of the general understanding of fibrosis described in IPF and some similarities between mechanisms leading to the IPF and LC progression. The manuscript is well written, and the structure of the review is well designed. I have only a few minor suggestions for improving the paper.
Comment 1: I suggest that the Authors should add a short section about CAFs in lung cancers, because despite similar mechanisms the most sections of this article describes IPF. It would be nice, if the Authors described the mechanisms of TGF-beta / Smad and TGF-beta / non-Smad signalling and mechanical factors in CAFs associated with lung cancer, not generally in CAFs.
Comment 2: Can the Authors explain if there are any reports about the role of histone deacetylases (HDACs) in the progression of IPF and LC as an element of epigenetic mechanisms of fibrosis?

Author Response

On behalf of all the Authors we would like to thank all the Reviewers for their valuable time and useful contribution to our Manuscript “Molecular mechanisms and cellular contribution from lung fibrosis to lung cancer development”. We strongly appreciate the suggestions and inputs they have given that will definitely improve our manuscript updating the molecular mechanisms that characterize both IPF and lung cancer onset and progression. Following their suggestions we have reorganized the manuscript to better and clearly explained concepts such as the role of Cancer Associated Fibroblasts (CAFs) particularly in cancer biology as well as the description of the current used drugs for IPF and LC. Please find below the details.

This review paper discusses the molecular mechanisms and cellular contribution from lung fibrosis to lung cancer development. Although the Authors postulated that the correlation between IPF and lung cancer is still being debated, in this review presented a lot of news and reports indicating the association of IPF with lung cancer development. This review gives a summary of the general understanding of fibrosis described in IPF and some similarities between mechanisms leading to the IPF and LC progression. The manuscript is well written, and the structure of the review is well designed. I have only a few minor suggestions for improving the paper.
Comment 1: I suggest that the Authors should add a short section about CAFs in lung cancers, because despite similar mechanisms the most sections of this article describes IPF. It would be nice, if the Authors described the mechanisms of TGF-beta / Smad and TGF-beta / non-Smad signalling and mechanical factors in CAFs associated with lung cancer, not generally in CAFs.

1)Agreed. We thank the Reviewer 3 for the suggestion that will definitively improve the manuscript impact. We have added the new section 6.1 entitled “The role of CAFs in tumor progression” on page 12-13 describes the role and the pathways characterizing the CAFs in tumors, not having a particular focus on lung cancer; this is because both the molecular mechanisms driving the activation of CAFs and their role in carcinogenesis may be common to various cancers and the majority of the literature describe the role of CAFs in other type of cancer such as pancreatic, breast etc..

“6.1 The role of CAFs in tumor progression

Besides fibrotic process, the paracrine activity of myofibroblasts which is carried out with cytokines and growth factors secretion, can affect tumor proliferation and progression, immunosuppression, angiogenesis and vascular remodelling (227, 228). Cancer associated fibroblasts (CAFs), similarly to myofibroblasts in IPF, are key components of the tumour microenvironment, (229, 230), whose recruitment and activation is mainly governed by the cytokines released by cancer cells and infiltrated immune cells. They originate from different cell populations as illustrate in figure 2 and explained in the Introduction. CAFs which represent the predominant cellular component of the tumour stroma, strongly contribute to the biology of tumor being essential in the mechanisms of the proliferation, invasion, inflammation, angiogenesis, and metastasis that take place in tumor microenvironment (TME). Furthermore, tumor cells and non-malignant stromal cells secrete different growth factors such as TGF-β1, EGF, vascular endothelial growth factor (VEGF), FGF, TNF-α, IL-1β and IL-6 (231, 232) that promote CAFs trans differentiation and activation contributing to a pro-inflammatory profile and carcinogenesis. Considering all, it is possible to note that there is a similarity between fibrosis and LC, and among the action of the different growth factors, TGF- β represents the key driver signaling for the trans-differentiation of both fibroblasts and CAFs, which are essential in tumor progression and resistance to therapies (233, 234). In recent years, many in vitro studies showed that TGF-β induced EMT into non tumor epithelial cells though the activation of mTOR and PI3K pathways, leading to apoptosis resistance of cancer cells, and thereby promoting the trans-differentiation of CAFs (235). Furthermore, CAFs are characterised by high heterogeneity, probably due to the exposure to different tumor secreted factors (TSF), that make them express different molecular markers. Despite their heterogeneity, the molecular mechanisms driving the activation of CAFs and their role in carcinogenesis may be common to various cancers. To this purpose, the main molecular pathways involved besides the TGF-β are the Wnt/β-catenin, the epidermal growth factor receptor, JAK/STAT, and Hippo pathways (236). Indeed, it has been shown that the different subgroups of CAFs can display opposite biological functions in cancer biology given the identification of some tumor-suppressive CAF populations that are characterized by activated Hedgehog signaling pathways in mouse models of colon, pancreatic, and bladder cancers. (237). Furthermore, it has been demonstrated the depletion of CAFs in a mouse model of pancreatic ductal adenocarcinoma (PDAC), caused hypoxia, cancer cell proliferation, and cancer progression revealing the tumor-suppressing role of CAFs at this level. (238). On the other side, Duda et al; (239) demonstrated in a preclinical studies of lung cancer that the ablation of CAFs reduced the number of metastases. Indeed, the authors after analyzing human brain metastases from lung, breast, kidney, and endometrium, they found high expression of activated CAFs within these metastases. These results demonstrated that the CAFs from the primary tumor might migrate in blood circulation proliferating at the metastatic site. Finally, CAFs contribute to the increased stiffness of tumour, that lead to the impairment of the blood vessels function, resulting in hypoxia that cause the inefficient uptake of anti-cancer drugs (234; 240). In conclusion, given the pivotal role of myofibroblasts in tumorigenesis, they could represents promising strategy both to prevent organ fibrosis and to limit tumor progression.

Comment 2: Can the Authors explain if there are any reports about the role of histone deacetylases (HDACs) in the progression of IPF and LC as an element of epigenetic mechanisms of fibrosis?

3)We thank the Reviewer 3 for the question and we added details on the role of HDACs in IPF and LC on page 16 line 738.

Indeed, a preclinical study demonstrated the anti-fibrotic effects of CG-745, a histone deacetylase (HDAC) inhibitor, in Bleomycin Mouse Models. They demonstrated that CG-745 prevented collagen sintesis, inflammatory cell accumulation, and cytokines release. The anti-fibrotic effects of CG-745 may suggest a potential therapeutic effect of CG-745 on lung fibrosis (281). Pre-clinical studies demonstrated that increased expression of HDAC-1 was associated with lung cancer progression. Consequently, treatment with HDAC inhibitors (HDACi) has shown anti-proliferative activity in non-small cell lung cancer (NSCLC) cell lines. Although this promising results in pre-clinical studies, HDACi showed only modest efficacy in lung cancer clinical trials (282)
